Diversified caching algorithm with cooperation between edge servers

Sang Yongxuan 1
Guo Yukang 1
http://orcid.org/0000-0003-3598-5359 Wang Bo 1 wangb@zzuli.edu.cn
Song Ying 2
1 Software Engineering College, Zhengzhou University of Light Industry , Zhengzhou, Henan , China
2 School of Computer Science, Beijing Information Science and Technology University , Beijing , China
Somani Arun
Electronic publication date: 2025 Apr 30
Publication date: 2025
Volume: 11
Electronic Location ID: e2824
Received 2024 Dec 4; Accepted 2025 Mar 23
Copyright: © 2025 Sang et al.
Copyright year: 2025
Copyright holder: Sang et al.
License: This is an open access article distributed under the terms of the Creative Commons Attribution License, which permits unrestricted use, distribution, reproduction and adaptation in any medium and for any purpose provided that it is properly attributed. For attribution, the original author(s), title, publication source (PeerJ Computer Science) and either DOI or URL of the article must be cited.
License URL: https://creativecommons.org/licenses/by/4.0/

Keywords: Edge caching, Edge computing, Edge cooperation, Edge cloud

Funding: National Natural Science Foundation of China 62102372 Project of Science and Technology in Henan Province 252102211072 and 232102210078 Doctor Scientific Research Fund of Zhengzhou University of Light Industry 2021BSJJ029 School-Level Young Backbone Teacher Training Program of Zhengzhou University of Light Industry The research was supported by the National Natural Science Foundation of China under Grant No. 62102372, the Project of Science and Technology in Henan Province under Grant Nos. 252102211072 and 232102210078, the Doctor Scientific Research Fund of Zhengzhou University of Light Industry under Grant No. 2021BSJJ029, and the School-Level Young Backbone Teacher Training Program of Zhengzhou University of Light Industry. The funders had no role in study design, data collection and analysis, decision to publish, or preparation of the manuscript.

==============================
Edge computing makes up for the high latency of the central cloud network by deploying server resources in close proximity to users. The storage and other resources configured by edge servers are limited, and a reasonable cache replacement strategy is conducive to improving the cache hit ratio of edge services, thereby reducing service latency and enhancing service quality. The spatiotemporal correlation of user service request distribution brings opportunities and challenges to edge service caching. The collaboration between edge servers is often ignored in the existing research work for caching decisions, which can easily lead to a low edge cache hit rate, thereby reducing the efficiency of edge resource use and service quality. Therefore, this article proposes a diversified caching method to ensure the diversity of edge cache services, utilizing inter-server collaboration to enhance the cache hit rate. After the service request reaches the server, if it misses, the proposed algorithm will judge whether the neighbor node can provide services through the cache information of the neighbor node, and then the server and the neighbor node jointly decide how to cache the service. At the same time, the performance of the proposed diversified caching method is evaluated through a large number of simulation experiments, and the experimental results show that the proposed method can improve the cache hit rate by 27.01–37.43%, reduce the average service delay by 25.57–30.68%, and with the change of the scale of the edge computing platform, the proposed method can maintain good performance.

Introduction

In the current context of rapid development in the mobile internet industry, with the global mobile data traffic forecast indicating that the number of Internet of Things (IoT)-connected devices in the world will exceed the population of the Earth. Low latency is of great significance for network transmission, as evidenced by the case of online surgery (Nunna et al., 2015). The concept of the Internet of Everything (IoE) is currently a topic of considerable interest and debate, and is playing an increasingly pivotal role in the development of modern intelligent applications (Zhou et al., 2023). The rapid development of the IoT has given rise to numerous compute-intensive and latency-sensitive emerging applications (Chen & Hao, 2018). In order to satisfy the requisite specifications of low latency, mobile cloud computing proposes the offloading of these application services to the cloud centre. However, the cloud centre not only has unpredictable wide area network delay, but also because the physical location is far away from the user, it is difficult to ensure the service quality of more and more application services that are sensitive to time delay. In response to this challenge, mobile edge computing has emerged as a solution. Compared with cloud computing, mobile edge computing is geographically closer to the user and has less network latency (Shi et al., 2016). The use of caching technology, whereby content is stored in devices at the edge of the network for future use, has the potential to significantly reduce backhaul traffic, a considerable proportion of which is the result of the repeated transmission of popular content to multiple users (Ahmed, Ameen & Zeebaree, 2021). Edge caching has the ability to considerably diminish the volume of redundant traffic. Conversely, the edge server has a limited storage capacity, which restricts the number of application services that can be cached (Xu et al., 2022). It is therefore essential to employ an appropriate caching algorithm to ensure that the application services stored on the server can be accessed by users as frequently as possible. The term ‘service caching’ is used to describe the process of storing application services and associated data on a base station or edge cloud (Taleb et al., 2017; Pan & McElhannon, 2018), with the aim of enabling the corresponding service to be executed at the edge (Xu, Chen & Zhou, 2018). Each time an edge server provides service to a user, cache hits enable the server to directly deliver the service, thereby reducing communication between servers and significantly lowering service latency. In the case of a cache miss, inter-server communication is required to determine the service provisioning approach. If the edge server is unable to provide the service, communication between the edge and the cloud is necessary, with the cloud server ultimately delivering the service. This process incurs substantially higher service latency. It is a challenging task to determine the optimal customisation of the caching scheme strategy in order to achieve the greatest possible resource utilisation (Li et al., 2024). Some caching schemes fail to consider the potential for collaboration between edge servers (Xu, Chen & Zhou, 2018), the capacity of edge servers (Yuan, Sun & Lou, 2022; Chen et al., 2018), or the evolving nature of user requests (Xu & Li, 2024).

In order to develop a reasonable caching strategy, this study focuses on the cache hit ratio, which has a direct impact on the service delay. A cache hit can be classified as either direct or indirect. A direct hit does not necessitate communication between edge servers and is characterised by lower latency. However, a focus on direct hits alone may result in a reduction in indirect hits, thereby increasing the miss hit rate, increasing communication between edge and cloud servers, increasing pressure on the cloud server and reducing service quality. Consequently, this study aims to achieve a balance between the overall hit rate and the direct hit rate, with the overall hit rate considered the primary factor, followed by the maximisation of the direct hit rate, in order to design the cache algorithm.

Related work

Some methodologies employ the conventional standalone memory cache replacement approach in edge cloud contexts. This includes the least frequently used page replacement algorithm (LFU) (Ko et al., 2022; Ascigil et al., 2022) and the least recently used page replacement algorithm (LRU) (Wang et al., 2020). These traditional algorithms do not entail iterative updates, exhibit low time complexity, and demonstrate robust performance on a single machine. The direct application of traditional algorithms to edge cloud environments, which are characterised by high dynamism (Ma et al., 2020), typically results in suboptimal performance. Literature (Zhang et al., 2021) posited that the cache algorithm would be optimally designed based on the assumption that end users have similar information needs when their information is similar. The cache hit rate is high, and positive outcomes have been observed in both service caching and service unloading. This has effectively reduced the overall service execution delay and achieved load balancing among devices. However, it is not responsive to changes in the information itself, and the cache information must be updated after each time slot. The literature (Yuan, Sun & Lou, 2022) proposes a cache algorithm for dynamic virtual edge nodes based on hierarchical clustering. This algorithm is capable of locating virtual edge nodes and actively allocating corresponding service resources with low time complexity. Literature (Chen et al., 2018) addresses the constraints of service quality and reduces user resource occupation by determining the optimal communication and computing resource curve of users. It should be noted that the proposed edge-cache algorithm does not consider the cache capacity of each edge node. Literature (Xu, Chen & Zhou, 2018) addresses the issue of dynamic service caching in the context of dense cellular networks. Nevertheless, this article solely contemplates the collaboration between edge servers and cloud servers, neglecting the potential for collaboration with nearby edge servers. Literature (Farhadi et al., 2021) proposes a polynomial-time service caching algorithm that considers storage, communication, computation and budget constraints. However, this approach to service caching is not flexible enough for long-term optimisation. The behaviour of humans is closely related and predictable. Prior to the arrival of the request, popular objects can be pre-cached through the utilisation of artificial intelligence and big data analysis (Qi et al., 2023; Zhou et al., 2021; Wu et al., 2023). Literature (Li et al., 2022) integrates the popularity of services with user interest, proposes a dynamic interest capture model to mine individual user interest, and studies group interest on this basis, with the objective of determining the content of the cache. However, the relationship between service popularity and user interest is not considered, particularly the impact of fluctuations in user interest on service popularity. Literature (Wang et al., 2019) proposed the “In-Edge AI” framework, which integrates deep reinforcement learning and federated learning into mobile edge systems to optimize computing, caching, and communication, demonstrating near-optimal performance with low learning costs and strong adaptability while discussing future challenges in edge artificial intelligence (AI). Literature (Deng et al., 2024) addressed edge caching in 5G networks by formulating the problem as a mixed-integer nonlinear program and proposing a Deep Q-Networks (DQN)-based optimization method, which effectively reduces system energy consumption and backhaul traffic while achieving higher cache hit rates compared to traditional approaches. Literature (Bernardini, Silverston & Festor, 2013) introduced the Most Popular Content (MPC) caching strategy for Content-Centric Networking (CCN), focusing on caching only popular content to reduce cache volume while improving cache hit rates, thereby enhancing caching efficiency through extensive simulations. Literature (Qian et al., 2020) examines the design of joint prefetching and caching strategies in Mobile Edge Computing (MEC) networks, proposing a method based on Hierarchical Reinforcement Learning (HRL). The authors model the problem as a Markov Decision Process (MDP) and apply a divide-and-conquer strategy to decompose it into two subproblems: user cache optimization and base station cache optimization. These subproblems are addressed using Q-learning and Deep Q-Networks (DQN), respectively. By incorporating both the long-term popularity and short-term temporal correlations of user requests, the proposed method effectively predicts future user demands and optimizes bandwidth utilization, demonstrating excellent performance across various scenarios. In literature (Moayeri & Momeni, 2024), an Edge Intelligent Context-aware Caching (EICC) system is proposed, which leverages environmental and user contexts to grade content through a content rating algorithm. This approach optimizes the selection of cached content, thereby enhancing prediction accuracy, cache hit ratio, and reducing latency. Literature (Kardjadja et al., 2024) introduces a deep reinforcement learning (DRL)-based strategy aimed at optimizing the caching of IoT edge gateways through intelligent content placement, taking into account content popularity and data diversity. Experimental results demonstrate that this method reduces data retrieval distance, enhances energy efficiency, decreases latency, and exhibits superior performance in terms of cache hit ratio, content diversity, and content popularity. In literature (Li et al., 2024), a 6G-based large-scale collaborative mobile edge computing (MEC) architecture is presented, which aims to improve the cache hit ratio for user requests through fine-grained contextual caching optimization. The architecture employs a LinUCB-based (Li et al., 2010) caching optimization algorithm to address the identified issues. Literature (Jin et al., 2024) proposes a vehicular edge network model targeting the optimization of average transmission delay by adopting a Centralized Edge Collaborative Caching (CECC) strategy, transforming the problem into a multiple-choice knapsack (MCK) problem. By designing a greedy algorithm, an approximate optimal solution is obtained to optimize the selection of cached content, effectively enhancing cache performance and reducing transmission delays. These studies collectively underscore the significance of edge caching within the realm of edge computing, illustrating that rational caching strategies play a pivotal role in enhancing user experience.

The server collaboration, whether service popularity changes, cache update cycle, and server cache capacity constraints discussed in the referenced literature are summarized in Table 1. In our study, we consider the collaboration among edge servers. To simulate a more realistic scenario, we assume that service popularity varies over time, and each server has a strictly limited caching capacity. To ensure rapid responses to services with sudden increases in popularity, our research focuses on achieving low cache computation time complexity and short cache update cycles. Each service request to the edge may potentially lead to changes in the entire service network’s cache. This article addresses the service caching optimization problem in mobile edge computing under dynamic service caching demands by leveraging the collaborative characteristics of multiple edge devices. We improve upon the First-In-First-Out (FIFO) and Least Recently Used (LRU) page replacement algorithms, proposing a diversified caching strategy. The goal is to maximize the variety of cached services to enhance the overall cache hit rate, thereby reducing the average service execution latency. Experimental results demonstrate that the proposed algorithm significantly decreases the average service completion latency and improves user experience quality.

Table 1 Partial constraints in referenced literature.

	Server collaboration	Service popularity	Cache update cycle	Cache capacity	
Literature (Zhang et al., 2021)	Yes	Unchanged	Long	Limited	
Literature (Yuan, Sun & Lou, 2022)	Yes	Unchanged	Short	Unlimited	
Literature (Chen et al., 2018)	Yes	Unchanged	Long	Unlimited	
Literature (Xu, Chen & Zhou, 2018)	No	Unchanged	Long	Limited	
Literature (Farhadi et al., 2021)	Yes	Unchanged	Long	Limited	
Literature (Li et al., 2022)	No	Change	Long	Limited	
Literature (Wang et al., 2019)	Yes	Unchanged	Long	Limited	
Literature (Deng et al., 2024)	No	Unchanged	Long	Limited	
Literature (Bernardini, Silverston & Festor, 2013)	No	Unchanged	Long	Limited	
Literature (Qian et al., 2020)	Yes	Changed	Long	Limited	

Materials and Methods

The system model presented in this article adopts an edge-cloud collaborative architecture, as illustrated in Fig. 1. The edge-server bears the responsibility of accepting and processing requests from end users. Furthermore, in the event that the edge server is unable to serve the requested action, it will forward the request to the cloud server, which will then serve on behalf of the edge server. However, it should be noted that the edge server is constrained by limited storage and computing resources. In order to optimise the utilisation of edge servers’ resources, this article proposes a collaborative strategy in which the communication delay between edge servers is reduced in comparison to that between edge clouds (Nouhas, Belangour & Nassar, 2023). In order to alleviate the burden on cloud servers, in the event of an unserviceable request, the edge server initiates a forwarding request to the neighbouring edge server, and subsequently transmits the request to the cloud server in the event that the neighbouring edge server is unable to fulfil the request. The objective of this article is to determine the optimal configuration for the service cache, with the aim of minimising system latency and enhancing the ability to satisfy service requests with low latency.

Figure 1 System model.

User request model

The collection of all service types is designated as S={1,2,...,s}. Each service necessitates the allocation of resources for storage. Consequently, the edge server cache service type s requires a specific quantity of storage resources cs. In practical terms, each request is processed at different times, although the network is utilized for disparate types of content. Nevertheless, only a modest proportion of the frequently required resources are typically necessary (Sun & Ansari, 2018), hence the potential benefit of cache hot content on the edge server in reducing the redundant transfer from the remote server (Wang et al., 2014). In this article, the probability of the service being executed is defined as the fitness value. A higher fitness value indicates a greater ease of selection for the service in question. The greater the popularity, the greater the fitness, and the service fitness is indicated by the symbol fs. In order to create a more realistic simulation, the fitness value is subject to change dynamically (Tang et al., 2024), with the possibility of a transition from a high to a low value. This may result in the server discarding the cache of the service, or vice versa. In the event that the caching algorithm is not sensitive to the perception of fitness, it may result in less fitness services retaining a large cache on the server, or conversely, services with higher fitness levels not having a cache on the server, which in turn may affect the hit rate. The symbol rf represents the rate of change of service fitness. On each occasion that the server provides a service, there is a probability of rf randomly selecting a service to change its fitness. The variables represented by each symbol are shown in Table 2.

Table 2 Variable symbols.

Variables	Description	
Nh	Number of direct hits	
Ni	Number of indirect hits	
Nm	Number of misses	
Rh	Direct hit rate	
Ri	Indirect hit rate	
Rm	Misses rate	
Wh	Directly hits the delay weight	
Wi	Indirect hit delay weight	
Wm	Missed delay weight	
T	Total delay	
S	Collection of all service types	
cs	Storage resources required for the service s	
fs	Fitness of the service s	
N	Collection of all of the servers	
n+1	Cloud server	
Mn	Collection of users served by the server n	
M	Collection of all of the users	
xsn	Whether the service s is stored on the server n	
Sn	Collection of services cached by server n	
mn	The server n provides the service to the users	
smn	Service requested by the user mn	
Rn	Storage capacity of the server n	
rf	Rate of change in service fitness	
Th	Direct hit delay	
Ti	Indirect hit delay	
Tm	Missed delay	

Server model

It is assumed that each edge server determines its optimal deployment location through optimization algorithms, and every service node employs load-balancing mechanisms to handle an equal amount of concurrent requests and service tasks. The server set comprises N edge servers and a cloud server, represented by n+1, and the set of all servers is N={1,2,3,...,n,n+1}. It is necessary for each edge device to provide a cache service for Mn users, the collection of users served by server n, which is Mn={1n,2n,3n,...,mn}, and the set of all users, which is M={M1,M2,M3,...,Mn}. The decision regarding the caching of services is expressed as xsn∈{0,1}, which indicates whether the service s is cached on server n, with a cache value of 1, otherwise 0. The remote cloud server storage cache contains all services, while the edge server, with limited storage resources, can only cache some resources. The service cache can be downloaded from the cloud server and adjacent edge servers. The storage capacity of server n is expressed in Rn.

The service storage constraint can be expressed as follows:

(1) ∑s⁡csxsn≤Rn,∀n∈N.

Service processing model

The service set of the server n cache is represented as Sn, and the user unloads the requisite service to the edge, where it is subsequently returned to the user. Given that the edge server is unable to store all the requisite services, it follows that the higher the success rate of the uninstallation service, the smaller the overall delay in service completion in this scenario.

The service requested by the user mn is expressed as smn. If smn∈Sn, this indicates that the service has been loaded into the edge server’s cache and is therefore directly accessible. If smn∉Sn and smn∈{Si|i∈Nandi≠n+1}, this signifies that the service has been loaded into the unloaded edge server’s cache but not the other edge server’s cache, and is therefore indirectly accessible. If smn∉{Si|i∈Nandi≠n+1}, this indicates that the service has not been loaded into any edge server’s cache, and is therefore unavailable.

The mean time to completion of the service can be calculated by weighing each hit rate and the corresponding specific proportion. The mean time to completion of the service can be expressed as follows:

(2) T=Wh∗Rh+Wi∗Ri+Wm∗Rm.

The variables indicated by the symbols are presented in Table 2 for reference.

(3) Rh+Ri+Rm=1.

The parameter values are presented in Table 3 for reference.

Table 3 Simulation parameters.

Parameter	Value	
Directly hits the delay weight Wh	1	
Indirect hit delay weight Wi	3	
Missed delay weight Wm	10	
Storage size cs of the service type s	[1, 20] GB	
Fitness fs of the service type s	[1, 200]	
Storage capacity Rn of the server n	500 GB	
All service types s	500	
Number of users mn of the server n	500	
Rate of change in service fitness rf	20%	
Number of edge servers	[8, 19]	

The diversified caching algorithm is based on the interconnection between servers, with the objective of maximising the variety of service caches. The LRU and FIFO algorithms demonstrate favourable performance and low time complexity in single machine replacement algorithms. However, their efficacy is limited in scenarios involving server cooperation. In light of the favourable outcomes observed for the LRU and FIFO algorithms, and in consideration of the attributes inherent to a diversified cache strategy, improvements have been made to these algorithms. In instances where a service cache is not directly accessed, the LRU and FIFO algorithms are modified to allow for selective abandonment of the service cache. By enabling neighbouring servers to cooperate, it is possible to ensure a greater variety of service types, increase the overall hit rate of services, and reduce service delays. The quality of the service provided is to be enhanced.

LRU algorithm

The LRU algorithm maintains a sequence table, which is used to record the order in which elements are processed. Upon the introduction of a new element, the list header and the original element will be removed from the end of the list. In the event that the newly inserted element is already present within the list, the existing elements will be advanced to the position of the list header. The data structure is subject to frequent queries and modifications, which are conducted through the list in order to construct a sequence table. The time complexity of each modification operation is o(1), as is the time complexity of the construction of a hash table for the purpose of a query. The time complexity of a query is also o(1). Each node of the list and hash table contains the address of the other, thus establishing a link between the time complexities of the query and the construction of the hash table. Consequently, the time complexity of each operation involving a new element is also o(1).

FIFO algorithm

The FIFO algorithm maintains a sequence table, which is used to record the order in which items are processed. Upon the addition of a new element to the list header, the original element is removed from the end of the list. In the event that the newly inserted element is already present within the list, there is no requirement for it to be inserted again. The data structure is subject to frequent queries and modifications, which are conducted through a queue in order to build the sequence table. The time complexity of each modification operation is o(1), while the time complexity of building a hash table for querying is also o(1). Furthermore, each node in the queue and hash table stores the other address, resulting in a time complexity of o(1) for mutual queries between the queue and hash table. Consequently, the time complexity for inserting new elements is also o(1).

Diversified caching algorithm

The evaluation standard for the experiment is the success rate of the service and the average time taken to complete the service. Table 4 details the process of the caching algorithm.

Table 4 Diversified caching algorithm.

Algorithm 1: Diversified caching algorithm	
The User mn sends the service smn to the Edge server n	
if ( smn∈Sn) do	
         Nh += 1;	
else if ( smn∈{Si|i∈Nandi≠n+1}) do	
         Ni += 1;	
        if ( ∑s⁡csn+cs≤Rn) do	
                  Server cache of receiving services the received service	
        end if;	
else do	
         Nm += 1;	
        if ( ∑s⁡csn+cs≤Rn) do	
                  Server cache of receiving services the received service	
        else if ( ∃∑s⁡csn+cs≤Rn) do	
                  The service is forwarded to a server with sufficient space	
                  and is hosted by a server with sufficient space	
        else do	
                  Server cache of receiving services the received service	
        endif	
endif	
The Edge server n returns the service smn to the user mn	

In this article, the edge devices employ orthogonal frequency division multiple access (OFDMA) for communication with terminal users. Specifically, each associated terminal user is assigned an orthogonal channel by the edge device for data transmission. As a result, interference between different transmission channels is effectively eliminated. When a user requests a service from the server, the server first checks whether the service is cached locally. The time complexity for querying the cache content of the current server can be expressed as follows:

(4) o(Th)=o(1).

If a cache miss occurs, the server sends a service request to adjacent servers. Since service caches are dynamically changing, each uncached service may impact the caching information within the server network. In this study, we adopted a multi-attribute decision making (MADM) approach, specifically utilizing the weighted Technique for Order Preference by Similarity to Ideal Solution (TOPSIS) algorithm, to optimize node communication performance in edge computing networks. By comprehensively considering factors such as distance, bandwidth, latency, load, and reliability, we selected the optimal neighbor for each edge node. Furthermore, dynamic cache changes may lead to frequent communication overhead; therefore, we incorporated a cache prefetching mechanism into our neighbor selection strategy to minimize unnecessary communications. Edge servers communicate using the WebSockets protocol to ensure timely data exchange and low-latency characteristics. The process of querying the cached content of other servers includes both request time and query time. The request time refers to the duration from sending the request until it is received by the neighboring node, and this value is incorporated into the calculation of the indirect hit proportion. The time complexity of indirect hits can be expressed as follows:

(5) o(Ti)=o(∑n⁡1)=o(n).

Concurrently, the edge server that is receiving the service will ascertain whether it is sufficient to cache the service without reducing the cache service, if there is adequate space available, satisfying Formula (6), it will proceed to cache the service. This could lead to an increased utilization of the edge server’s resources and an elevated direct hit rate for the service. To ensure that the range of services cached across all edge servers is as comprehensive as possible, caching will not occur if there is insufficient space, meaning the server does not meet the requirements of Formula (6). Should the original service be uninstalled while the new service is cached, the uninstalled service might become the sole data item cached across all edge servers. This reduction in the variety of service types available on the edge server would subsequently result in a decrease in the direct hit rate.

(6) ∑s⁡csn+cs≤Rn

In the event of a cache miss, if the requested service is also not cached by any of the adjacent servers, a total cache miss occurs, and the service request must be forwarded to the cloud server. The time complexity for this process can be expressed as follows:

(7) o(Tm)=max[o(Ti)]=o(n).

According to the caching algorithm, service resources should originally be cached by the server providing the service. However, in consideration of the diversity of service types, the diversified caching algorithm prioritizes identifying an edge server capable of caching the service. If such a server exists, as indicated by satisfying Eq. (8), the server providing the service will not cache the service, and instead, the service will be cached by the edge-end server with sufficient caching capability. This approach is consistent with the rationale for not caching services due to indirect hits. Conversely, if no such server exists, meaning Eq. (8) is not satisfied, the edge-end server receiving the service will offload existing cached services to make space for caching the new service.

(8) ∃∑s⁡csn+cs≤Rn

The time complexity of the algorithm

In the worst-case scenario, when there is a cache miss and none of the edge-end servers have the capacity to cache the service, it becomes necessary to perform service queries and space queries across all servers. The space query involves only a numerical comparison, resulting in a time complexity of o(1). Consequently, the time complexity of the algorithm can be expressed as:

(9) o(T)=max[o(Th),o(Ti),o(Tm)]+o(1)=o(n).

Results

In this section, simulation experiments are conducted to evaluate the performance of the diversified caching algorithms by varying the number of edge servers. Based on the parameters constrained in reference Xu & Li (2024), appropriate modifications are made, with each edge server serving 500 users. Since the number of servers is variable, an additional 500 users are added with each increase in the number of servers. The range of server numbers is incrementally set from eight to 19 to simulate algorithm performance under different server configurations. The remaining simulation parameters are provided in Table 3, where the proportion of the hit ratio is used solely for calculating service latency.

The results of the proposed algorithm are subjected to analysis. In order to evaluate the efficacy of the proposed algorithm, a comparison is made between the hit rate and response delay of the proposed algorithm and the traditional LRU and FIFO algorithms, respectively. Additionally, the performance of the traditional algorithm when it is online is also considered. To reduce the margin of error, each experiment was performed 100 times, and the mean value was calculated to enhance the precision of the experimental results.

The performance indicators are as follows: the direct hit rate of the service, the indirect hit rate of the service, the overall hit rate of the service, and the average delay of the service.

We compare our approach with recent works in the literature. Many state-of-the-art methods leverage reinforcement learning, which, however, demand extensive training data and computational resources, leaving room for improvement in terms of practicality. In our future work, we aim to integrate traditional single-machine caching strategies into reinforcement learning frameworks to enhance both the training efficiency and performance effectiveness. Furthermore, we plan to conduct a comparative analysis against the latest reinforcement learning-based caching strategies to validate the superior performance of our proposed method.

The effect of varying the number of servers on the overall hit rate is illustrated in Fig. 2. The overall hit rate of the proposed algorithm is consistently superior to that of the traditional algorithm. In the case of a limited number of servers, the total capacity of the edge servers is insufficient to accommodate the full range of potential services. Consequently, the proposed algorithm exhibits a notable increase in hit rate with an increase in the number of servers. When the total capacity of all edge servers is sufficient to accommodate all services, the edge will, in order to ensure the direct hit rate, abandon some services that are not often accessed. Consequently, the direct hit rate will only increase gradually with the addition of servers, and it is challenging to achieve 100%. The overall hit rate of the diversified LRU algorithm is 37.43% higher than that of the interconnected LRU algorithm when the number of servers is nine. In comparison to the interconnected FIFO algorithm, the overall hit rate of the diversified FIFO algorithm is increased by 27.01% when the number of servers is nine.

Figure 2 Effect of different number of servers on the overall hit rate.

The influence of disparate server numbers on the direct hit ratio is illustrated in Fig. 3. Irrespective of whether servers are connected or not, the direct hit ratio of the LRU algorithm is, in the majority of cases, slightly superior to that of the FIFO algorithm. Furthermore, with regard to the interconnection between servers, the algorithm with server interconnection demonstrates superior performance to the single algorithm in the majority of cases. As the number of servers increases, the total number of services that can be cached by the server also increases. Furthermore, the cache services of a server with a diversified caching algorithm tend to be stable. In order to ensure an overall hit rate, it is necessary to ensure that the cache services of each edge server are as different as possible. Consequently, the direct hit rate reflects a downward trend.

Figure 3 Effect of different number of servers on direct hit rate.

The impact of varying numbers of servers on the indirect hit rate is illustrated in Fig. 4. It can be observed that the standalone LRU and FIFO algorithms do not account for the interconnection between servers, resulting in the absence of indirect hits and a consistently zero indirect hit rate. As the number of servers increases, the service cacheable types also increase in the algorithms that consider the interconnection of the servers. Consequently, the algorithms’ indirect hit rates always increase with the increase in the number of servers. In this case, the diverse caching algorithm is based on the objective of maximising the total number of cached classes. The increase in the number of servers has resulted in a decrease in the direct hit rate, while the caching of a greater variety of services has led to a significant increase in the indirect hit rate. The indirect hit rates of the diversified FIFO and LRU algorithms are similar, and they consistently outperform the interconnected FIFO and LRU algorithms, which are optimal in terms of direct hit rates. Additionally, the indirect hit rate of the interconnected FIFO algorithm is higher than that of the interconnected LRU algorithm, which has a higher degree of randomness.

Figure 4 Effect of different number of servers on indirect hit rate.

The impact of varying server numbers on failure rates is illustrated in Fig. 5. It can be observed that failure rates consistently decline with rising overall hit rates. Conversely, failure rates tend to increase with the total service delay. Consequently, there is a positive correlation between failure rates and average delays. Furthermore, the trend of failure rates can, to some extent, reflect the trend of average delays.

Figure 5 Effect of different number of servers on misses rate.

The influence of varying server numbers on the mean service delay is illustrated in Fig. 6. In consideration of the interconnection among servers, both the FIFO and LRU algorithms are observed to reduce the average service delay as the number of servers increases. The service completion delay of the diversified caching algorithm with the highest overall hit rate is consistently lower than that of the traditional algorithm. The experiment demonstrates that it is possible to reduce the service delay by reducing the number of direct hits and increasing the number of indirect hits. The average latency of the diversified LRU algorithm is reduced by 30.68% when the number of servers is 11, in comparison to the interconnected LRU algorithm. A reduction of 25.57% in the average delay is observed when the number of servers is 12, in comparison with the interconnected FIFO algorithm.

Figure 6 Effect of different number of servers on the average execution latency of a task.

Conclusions

In the context of a limited number of edge servers, the diversified caching algorithm exhibits a markedly superior performance in comparison to alternative algorithms. As the number of edge servers increases, the total number and types of services that can be accommodated by edge servers also increase. Consequently, the advantage of the algorithm, which is to pursue the overall hit rate at the expense of the direct hit rate, gradually becomes less pronounced. In circumstances where the total number of services that can be accommodated by edge servers is significantly greater than the total number of existing services, the disadvantage of the algorithm is maximised, and the performance advantage is gradually reduced.

The proposed algorithm determines the minimum delay by improving the indirect hit ratio and fixing the delay formula. In order to guarantee the diversity of the cached services at the edge, the variety of service types will be increased, while the quantity of the same service in the cache is insufficient. In the event that the number of cached services is insufficient, the queuing delay will increase, resulting in subsequent services being blocked. This issue can be addressed by striking a balance between the diversity of service types and the number of caches of the same service. In the future, we aim to explore a hybrid approach that combines diversified caching with predictive models to address this issue.

In conclusion, the algorithm is applicable to scenarios where the total number of services is considerable and the total capacity of edge servers is not significantly greater than the total memory of the total number of services.

Supplemental Information

Supplemental Information 1 Caching hit rates of various methods.

Supplemental Information 2 Source code implemented by Python for the experiment data.

Additional Information and Declarations

Competing Interests

The authors declare that they have no competing interests.

Author Contributions

Yongxuan Sang conceived and designed the experiments, analyzed the data, prepared figures and/or tables, authored or reviewed drafts of the article, and approved the final draft.

Yukang Guo conceived and designed the experiments, performed the experiments, performed the computation work, prepared figures and/or tables, authored or reviewed drafts of the article, and approved the final draft.

Bo Wang performed the experiments, performed the computation work, authored or reviewed drafts of the article, and approved the final draft.

Ying Song performed the experiments, analyzed the data, authored or reviewed drafts of the article, and approved the final draft.

Data Availability

The following information was supplied regarding data availability:

The raw measurements are available in the Supplemental File.

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
