# Peer review of "Diversified caching algorithm with cooperation between edge servers"

_PeerJ Computer Science, doi:10.7717/peerj-cs.2824_

## Round 0.1 · original submission · Major Revisions

Please address all the comments of the reviewers in your revision

·

Basic reporting

1. The abstract requires significant improvement to meet publication standards. It should clearly and concisely present the background, motivation, the problem being addressed, the methodology used to solve it, the results obtained, and the key conclusions within 200-250 words. At present, the abstract fails to explicitly identify the problem or justify why it is critical, thereby diminishing its impact and clarity.
2. The language used in the paper should be revised to ensure it is precise and comprehensible to an international audience. Rather than relying on synonyms that may not accurately convey the intended meaning, select terms that clearly articulate the context and problem. For example, lines 23 and 37 should be rephrased to ensure clarity. Additionally, ensure proper formatting, such as including spaces before citations for better readability and alignment with publication standards.
3. The paper lacks an exhaustive discussion of related works and fails to reference key literature in the field.
4. The authors should clearly delineate how their work advances the state of the art. A dedicated related work section (e.g., Section II) is recommended to compare existing research and highlight the improvements introduced by this study. At the end of this section, a brief summary of the related work should be provided to emphasize the distinctions and contributions of the current research. A table summarizing related works and the gaps they leave unaddressed could serve as an effective tool to enhance clarity.
5. Figure 1, as presented, is overly simplistic and fails to encapsulate the intricacies of the problem. Additional details should be incorporated to make the figure more representative of the problem's nuances.
6. All figures must be self-explanatory. Ensure that each figure provides sufficient information to enable readers to fully grasp its relevance and significance without relying on the main text.

Experimental design

1. The "Experimental Results and Analysis" section must include detailed descriptions of the simulation setup. This includes specifying the parameters used for comparison, as well as the reference algorithms, to provide a clear context for readers to interpret the results.
2. To strengthen the manuscript, the authors should perform a comparative analysis by implementing and evaluating the algorithms proposed in other papers.

Validity of the findings

1. The restriction mentioned in lines 149-150 requires further explanation. The necessity of this constraint should be justified, especially in the context of a variable number of users and requests, which would make the scenario more realistic.
2. The service model should be formally defined using mathematical expressions to enhance rigor and clarity.
3. Algorithm 1 is difficult to comprehend due to the vertical lines inserted into the text. It should be rewritten for better readability and presentation.

Additional comments

Overall Comments: Several observations have already been addressed in the relevant sections for clarity and precision.

1. The concept of time-delay sensitivity must be clearly defined within the manuscript. Furthermore, the appropriate citation must be provided to substantiate the discussion. The citation referenced in line 56 (citation [8]) appears to be misapplied in line 60 and should be reevaluated or corrected. The statement, “this study focuses on the cache hit ratio, which has a direct impact on the service delay,” must be thoroughly justified by explaining how variations in the cache hit ratio influence service delay, as this is critical for validating the claim.
2. Given the focus on collaborative caching, the authors must elaborate on how cooperation among nodes is established and maintained. Additionally, the mechanism for identifying the cache status of neighboring nodes should be described in detail.
3. The problem scenario suggests that an edge node may fall within the communication range of multiple neighboring nodes. However, the manuscript does not provide a clear basis for selecting these neighbors. The criteria for such selection should be explicitly discussed.
4. The communication framework between the user and the edge server, as well as among edge servers and the broader internet, needs to be discussed comprehensively. These discussions are crucial to understanding the system's overall architecture and functionality.
5. In line number 119, the term "edge server" appears more appropriate than "cloud server." This discrepancy should be corrected for technical accuracy.

Cite this review as

Reviewer 2 ·

Basic reporting

- The manuscript is well-written, but there are occasional instances of ambiguous phrasing. A professional language review would be beneficial to ensure fluency and precision.

- The introduction provides a good contextual background but lacks a comprehensive discussion of content caching techniques like collaborative filtering. Including these would enrich the literature review and align the study with modern caching methodologies.

Experimental design

- The authors have clearly articulated the research gap and provided a novel solution. However, they could further emphasize the specific advantages of their approach over recent methods, particularly those involving machine learning-based caching.

- The proposed diversified caching algorithm is explained with sufficient detail, but the pseudocode and mathematical formulations could be clarified further to improve reproducibility.

- The simulation parameters and environment are well-defined. However, the authors should justify the chosen range of parameters (e.g., server capacity, service types) to ensure their applicability to real-world scenarios.

Validity of the findings

- The conclusion effectively summarizes the findings but lacks concrete suggestions for future work. A recommendation to explore hybrid approaches combining diversified caching with predictive models (e.g., reinforcement learning or collaborative filtering) could provide valuable direction.

- The provision of raw data is commendable, but additional explanation of how to replicate the experiments would be beneficial.

Additional comments

- The proposed diversified caching algorithm is explained with sufficient detail, but the pseudocode and mathematical formulations could be clarified further to improve reproducibility.

- The improvement in cache hit rates and service delay is significant, but additional metrics such as energy consumption and computational overhead could strengthen the study.

- Figures and tables are relevant and well-labeled, but the quality of some graphs (e.g., Figures 4 and 6) could be improved for better readability.

- The paper does not utilize collaborative filtering or similar techniques for content caching. Integrating such techniques could enhance the algorithm's adaptability to dynamic user behavior and content preferences.
Recommendation: Explore collaborative filtering to optimize cache content dynamically based on user interest and similarity patterns.

Cite this review as

---

## Round 0.2 · Minor Revisions

Please address the minor comments of the reviewer. Submit the revision with a note explaining what change you made. Thanks for interest in the journal.

Reviewer 2 ·

Basic reporting

Revised versions is better in basic reporting.

Experimental design

The designs are concrete and well defined.

Validity of the findings

Findings of the study are valid and well explained.

Additional comments

There is a minor revision. Some relevant and recent references to the area of content should be added. I only see one reference related to content caching that too is from 2014. As suggested earlier add 3-4 recent works to enhance the article visibility.

Cite this review as

---

## Round 0.3 · accepted · Accept

Thanks for revising the paper. It was a minimal change but hopefully makes the paper look more current and up-to-date on the literature review.